# Exploring the Mechanism of the Impact of Green Finance and Digital Economy on China’s Green Total Factor Productivity

**DOI:** 10.3390/ijerph192316303

**Published:** 2022-12-05

**Authors:** Jianfeng Guo, Kai Zhang, Kecheng Liu

**Affiliations:** 1Henley Business School, University of Reading, Berkshire RG9 3AU, UK; 2Economics and Management School, Xi’an University of Posts and Telecommunications, Xi’an 710061, China

**Keywords:** green finance, digital economy, green total factor productivity

## Abstract

In the context of the “double cycle,” promoting the development of a green economy is an important goal for China’s high-quality economic development in the digital age. This paper uses data from 30 provinces (municipalities and autonomous regions) in China during the 2006–2019 period using the Compiled Green Finance Index (GF) and Digital Economy Index (DE). The interrelationship between green finance, digital economy and green total factor productivity (GTFP) is empirically tested by conducting multiple regressions on panel data from 2006–2019 to perform an empirical analysis. Based on this, further analysis was performed with the threshold model. This study found that green finance and digital economy can contribute well to green total factor productivity, but the combination of the two does not have a good effect on green total factor productivity. Further study found that the green finance and digital economy’s contribution to green total factor productivity is mainly derived from technological progress. The regression results based on the panel threshold model show that the more underdeveloped the digital economy is in certain regions, the stronger the role of green finance in promoting efficiency improvement. Therefore, policymakers should formulate differentiated green financial policies according to the level of development of the digital economy and give play to the role of green finance and the digital economy in promoting green total factor productivity.

## 1. Introduction

Since the reform and opening up, China’s economy has been maintaining high growth, but the high economic growth stage has been accompanied by a dependence on resources and the pollution of the environment [1,2]. The report of the 19th National Congress of the Communist Party of China points out that “China’s economy has shifted from a stage of high-speed growth to a stage of high-quality development, and it is necessary to promote quality change, efficiency change, power change in economic development and improve total factor productivity.” In today’s recurring epidemic, the need to energize productivity growth is even more pronounced. As the concept of green development continues to penetrate the concept of national governance, how to improve the quality of the ecological environment and the development of the green economy has gradually become a greater concern for people [3,4,5]. Promoting GTFP and improving the efficiency of the green economy play important parts in promoting the development of China’s green economy [6,7]. The improvement of total factor productivity is mainly reflected in two aspects of technological progress and efficiency progress, and each general technological innovation in human history has been able to significantly promote the leapfrog development of these two aspects [8,9,10]. Green finance incorporates environment and pollution into endogenous factors, and through the development of related green credit and green investment business, it guides the flow of funds into green environmental protection projects, optimizes resource allocation, and achieves win-win interaction between green finance and green economy [11,12]. Therefore, green finance plays an important role in promoting high-quality economic development [13,14,15].

In recent years, thanks to the continuous breakthrough of information technology in China, the rapid development and wide application of digital technology have given rise to the digital economy. The digital economy is different from the traditional agricultural and industrial economies. As a new economy, it is deeply integrated with all industries in China, triggering great social and economic changes while providing a new path for China to achieve environmentally friendly and sustainable development because of its improved efficiency and reduced dependence on resources and the environment. The 14th Five-Year Plan of the National Economic and Social Development of the People’s Republic of China and the Outline of the Vision 2035 clearly proposed to “give full play to the advantages of massive data and rich application scenarios, promote the deep integration of digital technology and the real economy, empower the transformation and upgrading of traditional industries, give birth to new industries, new business models and grow new engines of economic development.” It is easy to see that the digital economy has become an important driving force for China’s economic development [16,17,18].

In the new era, China has the will and motivation to promote the two-pronged approach of green finance and digital economy to drive high-quality economic development. Green finance refers to all financial innovation and management activities that help achieve environmental improvement, enhance eco-efficiency and promote sustainable development functions. It includes not only environmental finance, low-carbon finance, and sustainable finance activities but also financial policies, financial services, risk management, and other related financial resource allocation activities adopted by governments, enterprises, and other economic agents that are conducive to promoting green investment and financing. Green finance is a financial innovation based on ecological and environmental protection, strengthening the link between the green industry and the financial industry, focusing on issues such as environmental pollution and ecological and environmental protection [19,20]. A digital economy is a new form of economic and social development after the agricultural and industrial economies. The G20 Initiative on Digital Economy Development and Cooperation, released at the 2016 Group of Twenty (G20) Summit, defines the digital economy as a series of economic activities in which the use of digitized knowledge and information is a key factor in production. Additionally, modern information networks are an important carrier of information, and the effective use of information and communication technologies is an important driving force for efficiency improvement and economic structure optimization. Thus, the digital economy has become an important engine for China’s high-quality economic development due to its efficient use of resources. GTFP is defined as the integration of input variables such as capital, energy and labor, economic benefits representing desired outputs and environmental pollution representing undesired outputs into the productivity measurement framework taking into account both increases in desired outputs and decreases in undesired outputs. We usually use GTFP as an indicator to measure and evaluate the quality of growth of an economy.

Based on the above background, in order to better propose countermeasures to promote high-quality economic development, this article estimates the level of green finance and digital economy by constructing a multidimensional indicator system and verifies the effects of green finance and digital economy on GTFP and its decomposition term using multiple regressions in a unified framework. We also use a threshold model to analyze the intensity of the impact of green finance and the digital economy on GTFP. This article first compares the existing relevant studies and then introduces the selection of variables and the setting of the model. Second, this article discusses the main sources driving GTFP by analyzing its decomposition term. On this basis, this article conducts an empirical analysis of green finance and digital economy acting on GTFP, its decomposition term separately, and green finance and digital economy acting on GTFP, and its decomposition term together. We discuss in depth the influence mechanism in the process of green finance and the digital economy affecting GTFP. After that, the threshold effects on the roles of green finance and digital economy in the decomposition term efficiency progress of GTFP are further investigated and discussed. Finally, the conclusions of this study are drawn, thus providing a theoretical basis for relevant policy formulation. The main conclusions provide not only new ideas for developing green finance, promoting the construction of the digital economy and enhancing GTFP, but also provide important references in the implementation of green development concepts for local governments.

## 2. Literature Review

Green finance, as a link between the financial and green industries, completes the measures to upgrade the industrial structure by means of financial support for the green industry to continuously improve technological innovation, in line with the law of energy development, transforming energy use from fossil to clean energy, optimizing fossil energy, strengthening the global energy transition, and promoting green development [21]. Green finance is a new financial innovation that combines the concept of finance with the green industry, which introduces financial market volatility and geopolitical uncertainty, but is generally beneficial to the development of green finance and the green industry [22]. At the same time, the dual strategic transformation underscores the undisputed complementary relationship between green finance and digital transformation [23]. Additionally, the digital economy has promoted the development of the green economy well due to its resource allocation optimization and technological innovation-driven industrial structure upgrading. Therefore, it is necessary to study the impact of green finance and the digital economy on the green economy in depth. A review of the available research results shows the following main aspects:(1)Research related to green finance and green economy.

Green finance is mainly through the guidance of financial institutions to make them invest in green projects that can bring energy saving and environmental protection to improve GTFP, as well as through social supervision to restrict the financial channels of high-polluting enterprises to either promote their transformation or green technology research and development, thus promoting GTFP. On the one hand, as the original energy-intensive production method is transformed into a green and environment-friendly production method, which has a very high cost, this requires green finance to provide financial support for green industries to optimize capital allocation [24,25,26]. In order to obtain more support from green loans, enterprises are more willing to take the initiative of carrying out the research and development of green technology and improve their own productivity. The incentivizing effect of green finance on enterprise technological innovation has well-promoted the development of GTFP [27,28,29]. By supporting green projects, green finance has greatly promoted environmental protection and played an important role in promoting China’s high-quality economic development [30]. On the other hand, green finance is a special fund used to promote green development projects. After assessing green finance projects and approving them for financial support from green financial services, enterprises that want to develop green projects have the obligation and responsibility to fulfill corresponding social and environmental protections. At this time, their production and operation behaviors need to be supervised by relevant supervision departments, and the funds they obtain through green finance channels need to be used in green-related industries, thus improving GTFP [31]. The impact of green finance on GTFP tends to show different effects in stages [32]. In the short term, the transformation of highly polluting industries and the establishment of new green industries often require large amounts of financial support. The long transformation cycle of highly polluting industries and the establishment of new green industries leads to high input and low output of green financial inputs, which will reduce GTFP, while in the later stage, the transformation of highly polluting industries and the establishment of new green industries bring output returns, which will increase GTFP. Therefore, green finance and GTFP often show a U-shaped fitting curve [33]. Secondly, due to its unique loan conditions, green finance will have a corresponding loan threshold when providing services to enterprises. After receiving the loan, enterprises will also be supervised by regulatory authorities to monitor whether the enterprise loan is used for green projects, so there is often a threshold effect in the process of promoting GTFP in green finance [34,35,36]. In order to obtain capital loans, enterprises need to purchase equipment and upgrade corresponding green technologies to meet the requirements of green development, which will increase the cost burden on enterprises in the short term [37]. As enterprises continue to expand their business, improve their production efficiency level, and meet the requirements of green development, they are able to obtain more financial support, be regulated by the corresponding regulatory authorities, pay more and more attention to green development, actively carry out green production, improve production efficiency, and thus increase the total green factor [38].

(2)Research related to digital and green economies.

As an emerging economy, the digital economy has a significant impact on the new information industry revolution, so the development of the digital economy puts forward new requirements for the policy system in the industrial economy era. On the one hand, the digital economy improves the efficiency of resource allocation through digital technologies, and this more efficient way of production contributes to GTFP [39,40]. It has been shown in the literature that the digital economy contributes to GTFP mainly through green technological change [41]. At the same time, the digital economy itself relies on network infrastructure and information tools, such as smart machines, which break the limitations of time and space through information technology and internet mode, giving human beings the ability to process big data and continuously disseminate a large amount of information. The development of this ability relies on continuous technological innovation and research and development, so there are financial thresholds as well as technical thresholds in the digital economy for GTFP development [42,43,44]. On the other hand, the digital economy can be deeply integrated into all walks of life by upgrading and positively impacting the transformation of the industrial structure [45,46]. In addition, the upgrading of the industrial structure has a significant impact on the improvement of GTFP [47,48]. As the development of China’s digital economy continues to promote China’s economy in a more equitable and efficient direction, the combination of traditional production industries and the digital economy tends to promote the flow of production factors from the primary industry to the secondary and tertiary industries, and the continuous optimization of resource allocation to more efficient sectors, effectively improving the degree of dependence of economic development on energy resources and promoting the transformation and upgrading of industrial structure to digitalization, rationalization, and greening [49,50]. Other literature has empirically tested the impact of the digital economy on GTFP at the provincial or city level, affirming a positive and significant impact of the digital economy on GTFP, but often with regional heterogeneity [51,52,53,54].

In summary, it can be seen that, although there have been rich discussions in the academic community about the impact of the digital economy and green finance on total factor productivity, the discussion on how to promote GTFP and the impact of the digital economy and green finance on GTFP in the context of new development concepts and digital economy is still insufficient, mainly in the following aspects. First, very little of the literature analyzes the impact of the digital economy and green finance on GTFP within the same framework and also ignores the specific sources of total factor productivity gains. Second, few studies have examined the mechanisms at play in the process of green finance and the digital economy affecting GTFP. Finally, the established literature is more concerned with analyzing the direct effects of green finance and the digital economy on the impact of GTFP, and is less concerned with the impacts of both technological progress and efficiency improvement.

## 3. Methods

### 3.1. Data Sources

The starting point of the study chosen for this paper is 2006, and the endpoint is 2019. The initial data of each indicator are mainly obtained from the statistical yearbooks of all Chinese provinces, China Statistical Yearbook, China Science and Technology Statistical Yearbook, China Energy Statistical Yearbook, CSMAR and CCER databases, China Foreign Direct Investment Statistical Bulletin, China Insurance Yearbook and China Industrial Statistical Yearbook, etc. Some of the missing data are filled in by linear interpolation. In addition, in the selection of inter-provincial samples, due to the problem of more missing data and inconsistent data caliber, this paper selected 30 provinces and cities, except for Tibet, Hong Kong, Macao and Taiwan, as the research subjects.

### 3.2. Indicator Setting

1. The explanatory variable, Green Total Factor Productivity (GTFP). Since Data Envelopment Analysis (DEA) has the advantage of not requiring functional assumptions, and the non-angle and non-radial distance of the Malmquist index (ML) can treat pollution emissions as a non-desired output and solve the problem of radial distance function, it can achieve a decrease in non-desired output and an increase in desired output at the same time. This paper draws on the measure of Chung [55] to measure the GTFP index using the DEA–SBM non-angle, non-radial distance Malmquist index. The input indicators in this paper include labor, capital, and energy, using the total number of employees at the end of the year to measure labor and capital input: $K_{i,t}=K_{i,t-1}(1-\delta_{i,t})+I_{i,t}$, where, K denotes physical capital stock, δ denotes depreciation rate (the value of δ was taken as δ = 9.6% by referring to Zhang [56]) and I denotes real fixed-asset investment in each province. Energy inputs are measured using society-wide electricity consumption, and desired output indicators are measured using the gross product. The entropy value method is applied to collate industrial wastewater emissions, industrial SO_2_ emissions, and industrial smoke (dust) emissions into a comprehensive index of environmental pollution to measure non-desired output indicators. The ML index can be further decomposed into technical efficiency change (EC) and technical progress change (TC). The specific expressions are as follows,
(1)$$ML_{t}^{t+1}=EC\times TC$$
(2)$$EC_{t}^{t+1}=\frac{1+\vec{D_{0}^{t}}\left(\chi^{t},y^{t},b^{t};y^{t},-b^{t}\right)}{1+\vec{D_{0}^{t+1}}\left(\chi^{t+1},y^{t+1},b^{t+1};y^{t+1},-b^{t+1}\right)}$$
(3)$$TC_{t}^{t+1}=\{\frac{\left[1+\vec{D_{0}^{t}}\left(\chi^{t+1},y^{t+1},b^{t+1};y^{t+1},-b^{t+1}\right)\right]}{\left[1+\vec{D_{0}^{t}}\left(\chi^{t},y^{t},b^{t};y^{t},-b^{t}\right)\right]}\times \frac{\left[1+\vec{D_{0}^{t+1}}\left(\chi^{t+1},y^{t+1},b^{t+1};y^{t+1},-b^{t+1}\right)\right]}{\left[1+\vec{D_{0}^{t+1}}\left(\chi^{t},y^{t},b^{t};y^{t},-b^{t}\right)\right]}{\}}^{\frac{1}{2}}$$
where EC means that a change in pure technical efficiency and a change in the efficiency of the scale of production causes a change in the internal efficiency of the producer, and the increase in industrial output resulting from this change is called a change in technical efficiency. TC means a change in industrial output caused by pure technological progress. The ML index is multiplied cumulatively to obtain the final GTFP.

2. Green Finance Index (GF). Some scholars have used green corporate bank loans, green credit share, green investment level, and green credit policy dummy variables as proxy variables for green finance. For the sake of comprehensiveness, this paper calculates provincial green finance development indicators using the composite index method based on data from 30 Chinese provinces and cities from 2006–2019. According to the definition of green finance, it mainly integrates four aspects: green credit, green investment, green insurance, and government support, among which green credit is the most important part of green finance, while other green financial products have gradually diversified in recent years, so green credit cannot be taken as the only indicator to measure the level of green finance. Referring to LY He’s [57] research ideas, while considering the validity and availability of data, the entropy value method is used to calculate the level of green finance in each province. The index system was constructed as shown in Table 1.

3. The Digital Economy (DE). The concept of the digital economy in economics refers to the use of big data, the rapid integration, optimization and regeneration of resources to achieve the optimal allocation of resources to achieve high-quality economic development from all digital integration of resources can be considered the digital economy, generally speaking. The digital economy is a major economic form after the development of agricultural and industrial economies. With modern information networks as the main carrier and data resources as the key element, it promotes the integration and application of modern information technology, facilitates modern digital transformation, changes people’s current life, production and governance, and is a new economic form that is more equitable and efficient. At present, the academic community continues to dig deeper into the digital economy as well as supplement and improve the evaluation system of digital economy indicators, mainly combined with infrastructure construction, internet level, and a series of elements to measure, compared with the previous single way of measurement methods, its measurement methods and levels continue to expand and deepen on the original basis but has not yet reached a unified standard. Nowadays, it can be determined that the core of the digital economy is digital resources, through modern information technology applications to provide consumers with convenient and fast services and products so that digital transactions become an emerging economic form of producers and consumers trading ties. Based on the existing literature and considering the availability and completeness of data, this paper constructs a measurement system containing four primary indicators and 27 secondary indicators, covering various elements such as digital infrastructure, digital penetration rate, digital technology talent benefits, and digital research. The data were mainly obtained from the China Statistical Yearbook, the Electronic Information Industry Statistical Bulletin and the provincial statistical yearbooks. Based on the construction of the index system, the KMO and the Bartlett test were conducted, and it was found that the KMO = 0.863 and the Bartlett test results proved that there were significant differences among the indicators. Principal component analysis can be used for dimensionality reduction. The construction of the index system is shown in Table 2.

4. Other variables. With reference to existing studies, the following control variables are selected in this paper: The level of openness to foreign investment (OPEN) is expressed as the share of total foreign direct investment in real terms in local GDP; The level of industrial structure (OIS) is expressed as the share of secondary industry output in local GDP; The level of urbanization (URB) is expressed as the share of the urban resident population within the resident population; Research, development, and investment (RD) is expressed as the number of local patents; Finally, the level of government spending (GOV) is expressed as a share of government fiscal spending in regional GDP. The above data are from the “China Statistical Yearbook,” “China Environmental Statistical Yearbook,” and the provincial statistical yearbooks.

### 3.3. Model Construction

In order to test the relationship between green finance, digital economy and GTFP, as well as the relationship between green finance, digital economy and the decomposition term of GTFP, and to deeply investigate the path of action on GTFP, Equations (4)–(6) are constructed in this paper.
(4)$${GTFP}_{it}=\alpha_{0}+\alpha_{1}{GF}_{it}+\alpha_{2}{DE}_{it}+\alpha_{i}\;{Controls}_{it}+\lambda_{i}+\mu_{it}+\varepsilon_{it}$$
(5)$${TC}_{it}=\beta_{0}+\beta_{1}{GF}_{it}+\beta_{2}{DE}_{it}+\beta_{i}\;{Controls}_{it}+\lambda_{i}+\mu_{it}+\varepsilon_{it}$$
(6)$${EC}_{it}=\gamma_{0}+\gamma_{1}{GF}_{it}+\gamma_{2}{DE}_{it}+\gamma_{i}\;{Controls}_{it}+\lambda_{i}+\mu_{it}+\varepsilon_{it}$$
where ${GTFP}_{it}$ denotes GTFP in province, i, in year, t, ${GF}_{it}$ is a measure of green finance, ${DE}_{it}$ is the digital economy, and ${Controls}_{it}$ represent control variables. ${TC}_{it}$ represents technological progress, ${EC}_{it}$ represents efficiency progress, $\lambda_{i}$ denotes time-fixed effects, $\mu_{it}$ denotes individual-fixed effects, and $\varepsilon_{it}$ is a random error term.

In order to clarify the mechanism of the interaction term between green finance and digital economy on GTFP and the effect of the interaction term between green finance and digital economy on the decomposition term of GTFP, the cross-product term ${GF}_{it}\times {DE}_{it}$ is introduced in the model to test the role played by the interaction term between green finance and digital economy in GTFP. The models constructed in this paper are Equations (7)–(9).
(7)$${GTFP}_{it}=\alpha_{0}+\alpha_{1}{GF}_{it}+\alpha_{2}{DE}_{it}+\alpha_{3}{GF}_{it}\times {DE}_{it}+\alpha_{i}\;{Controls}_{it}+\lambda_{i}+\mu_{it}+\varepsilon_{it}$$
(8)$${TC}_{it}=\beta_{0}+\beta_{1}{GF}_{it}+\beta_{2}{DE}_{it}+\beta_{3}{GF}_{it}\times {DE}_{it}+\beta_{i}\;{Controls}_{it}+\lambda_{i}+\mu_{it}+\varepsilon_{it}$$
(9)$${EC}_{it}=\gamma_{0}+\gamma_{1}{GF}_{it}+\gamma_{2}{DE}_{it}++\gamma_{3}{GF}_{it}\times {DE}_{it}+\gamma_{i}\;{Controls}_{it}+\lambda_{i}+\mu_{it}+\varepsilon_{it}$$
where ${GF}_{it}\times {DE}_{it}$ represents the cross-product term of green finance and digital economy. Finally, we verified whether the coefficient *α*_3_ in Equation (7), the coefficient *β*_3_ in Equation (8), and the coefficient *γ*_3_ in Equation (9) are significant.

## 4. Results

### 4.1. Descriptive Statistics and Correlation Analyses

The descriptive statistics of all analyses are listed in Table 3. The mean value of GTFP is 1.514, the maximum value is 4.979, and the minimum value is 0.608, which indicates a large difference in GTFP between regions. The minimum value of green finance development level is 0.050 and the maximum value is 0.0793. The minimum value of digital economy development level is 0.11 and the maximum value is 0.77. This indicates that the level of green finance development and the level of digital economy development in that there are also large differences between regions.

The development of GTFP with EC and TC in China from 2006–2019 is shown in Figure 1. As we can see in Figure 1a, the GTFP level increased significantly from 0.98 in 2006 to 2.45 in 2019. Comparing Figure 1b,c, it can be found that both technological progress and efficiency improvement have a significant increase from 2006–2019, but the increase in technological progress is closer to the GTFP improvement curve, which indicates that the green total factor improvement mainly comes from technological progress.

Green finance enables financial institutions to invest in green projects that can bring energy savings and environmental protection through the guidance of financial institutions. China’s economic transformation also promotes the concept of low carbon, energy saving, and environmental protection. For this reason, high-pollution enterprises face policy constraints as well as loan restrictions; Thus, they are more willing to comply with the concept of energy conservation and environmental protection through technological innovation and industrial structure innovation, which is conducive to raising the level of GTFP. The fitted graph of green finance and GTFP is shown in Figure 2a, which shows that there is a positive correlation between green finance and GTFP.

On the one hand, the development of the digital economy has freed the traditional economy from its heavy dependence on energy and the environment, as well as significantly reduced the excessive consumption of energy in the industrial economy model, improved the efficiency of factor utilization, and promoted green and high-quality development through energy conservation and emission reduction. On the other hand, the development of the digital economy has broken the geographical barriers and realized the cross-regional flow of talents, information, and technology, which is conducive to stimulating green technological innovation and improving GTFP. The fitted graph of the digital economy and GTFP is shown in Figure 2b, which shows that there is a positive correlation between the digital economy and GTFP.

The relationship between green finance and the digital economy is complementary. By combining with the high-energy digital economy industry, green finance can promote the related industries to continuously develop technology to reduce the energy consumption of the digital economy and help achieves low-carbon sustainable development. The digital economy, with its unique data and information technology, provides numerous benefits, such as a linked upstream and downstream platform for the green finance system, established information sharing and security mechanisms, greatly improved matching efficiency of investment and financing, the increased scale of green finance, and allows green finance to be well-integrated into the industry [10]. The fitted graph of green finance and the digital economy is shown in Figure 2c, which shows that there is a positive correlation between green finance and the digital economy.

Overall, green finance, the digital economy, and GTFP are generally on an upward trend. In terms of relationships, green finance–GTFP, digital economy–GTFP, and green finance–digital economy all have linear relationships.

This paper performs correlation test indicators, as shown in Table 4. The significant results between the independent and the dependent variables are extremely significant, which indicates that the independent variables selected in this paper are strongly correlated with the dependent variable. Secondly, the significance levels of the remaining variables have been tested, except for the level of foreign openness and the level of government expenditure, which are not significant.

In this paper, LLC tests were conducted on the independent and dependent variables, the significance levels were passed, and the test results are shown in Table 5.

### 4.2. Regression Analysis Results of Green Finance, Digital Economy, and GTFP

Column (1) of Table 6 shows the test results of Equation (1). It can be seen that the coefficient 2.115 of GF is significantly positive at the 1% level, which indicates that the development of GF contributes to the growth of GTFP, as it increases by 2.115% for every 1% increase in the level of GF. The coefficient of 1.991 for DE is significantly positive at the 1% level, which indicates that the development of the digital economy also contributes to the growth of GTFP, as it increases by 1.991% for every 1% increase in the level of development of the digital economy. The rest of the variables are economic variables essential to the operation of the economy. The coefficient of OIS −0.008 is significant at the 5% level, which indicates that the higher the share of secondary industry structure, the higher the inhibitory effect on GTFP. The coefficient of OPEN −4.426 is significant at the 1% level, which indicates that foreign investment has a suppressive effect on GTFP to some extent. The coefficient of URB 3.125 is significantly positive at the 1% level, which indicates that as urbanization continues, it can also contribute to the growth of GTFP. The coefficient of RD −40.160 is significant at the 1% level, which indicates that R&D and input inhibit the development of GTFP to some extent, which is due to the fact that enterprises are more inclined to deepen R&D on existing technologies in the process as they expect to achieve higher energy use efficiency in order to achieve the purpose of energy saving and environmental protection, rather than to develop new projects. This kind of R&D and input to improve energy efficiency rather than update to the green energy-saving industry as the purpose of R&D and investment, to a certain extent, caused the phenomenon of energy rebound, thus, it has a certain inhibitory effect on the growth of GTFP. Finally, GOV and GTFP are not significant at the level of significance.

Columns (2) and (3) are the Equations (2) and (3) constructed from the decomposition terms TC and EC of GTFP. It can be found that GF is significantly positive in both Equations (2) and (3). While DE is significantly positive in Equation (2), the coefficient is −0.416 and also significant in Equation (3). This suggests that GF increased both TC and EC, and contributed to the growth of GTFP by the coupling of the two. Finally, DE only increased TC and did not contribute well to EC, but in general, it also contributed to the growth of GTFP.

When combining Columns (1), (2) and (3), GF can increase the level of GTFP, and DE is able to improve GTFP. This is consistent with the findings of existing studies. However, the boost to GTFP comes mainly from TC.

Columns (4), (5), and (6) are the results after adding GF×DE.

The results of GF×DE in column (4) are not significant, and the results of GF are not significant. Only the coefficient of DE 1.930 is significantly positive at the 1% level. This suggests that the combined effect of GF and DE does not contribute well to the improvement of GTFP.

Columns (5) and (6) are the Equations (5) and (6) constructed for the decomposition terms TC and EC of GTFP. The coefficients of GF are −2.719, which is significant in Equation (5), and 2.867, which is significantly positive in Equation (6). The coefficient of DE is 2.005, which is significantly positive in Equation (5), but insignificant in Equation (6). The interaction of GF×DE have coefficients of 4.94, which is significantly positive in Equation (5), and −2.982, which is significant in Equation (6). This indicates that GF is able to produce significant positive effects when acting on GTFP as well as when EC acts on GTFP, but when GF is combined with DE, it has an inhibitory effect on EC during production. Since GTFP = EC×TC, the joint effects of GF and DE shows a significant increase in TC but also a significant inhibition in EC; Thus, the joint effects of GF and DE do not significantly increase GTFP.

The combined Columns (4), (5) and (6) show that GF and DE together have not been very good at significantly enhancing GTFP. Both of them were significantly positive when acting together on TC, but significantly negative when acting on EC. It is possible that GF and DE together did not achieve the expected results when acting on EC due to some limitation, which requires further analysis.

### 4.3. Threshold Effect

There is a complex relationship between GF, DE, and GTFP. On the one hand, GF helps GTFP by supporting green industries. On the other hand, DE is also able to improve GTFP by optimizing the allocation of resources. Although both of them can contribute to the improvement of GTFP in our country, they do not improve GTFP very well when there is a combined effect of the two. This “threshold effect” may exist in GTFP or in the decomposition of GTFP, TC, or EC, and it is because of this threshold effect that GF and DE together do not contribute well to the growth of GTFP. Based on this, this paper conducted threshold effect tests on GTFP and its decomposition terms TC and EC with GF and DE as threshold variables, respectively, and repeated the samples 300 times using the Bootstrap method. The results obtained are shown in Table 7.

As seen in row (1) of Table 7, the *p*-value of 0.6533 for the single threshold of GF to GTFP did not pass the significance test, thus concluding that there is no single threshold of GF to GTFP.

As seen in row (2) of Table 7, the *p*-value of 0.6533 for the single threshold of GF to TC did not pass the significance test, thus concluding that there is no single threshold of GF to TC.

From row (3) of Table 6, it can be seen that the *p*-value of 0.3033 for the single threshold of GF to EC does not pass the significance test, thus concluding that there is no single threshold of GF to EC.

As can be seen from row (4) of Table 7, the *p*-value of 0.6467 for the single threshold of DE to green GTFP did not pass the significance test, thus concluding that there is no single threshold for DE to GTFP.

From row (5) of Table 7, it can be seen that the *p*-value of 0.43 for the single threshold of DE to TC does not pass the significance test, thus concluding that there is no single threshold of DE to TC.

From row (6) of Table 7, it can be seen that the *p*-value of the single threshold of the decomposition term EC for DE on GTFP is 0.0367, which indicates that there is a threshold effect of DE on EC at the 5% significance level, while the *p*-value of the double threshold is 0.1033, which fails the test of a double threshold. Thus, there is a single threshold effect of DE on EC with a threshold value of 0.2020, and a single threshold effect is chosen. To verify the accuracy of the threshold estimates, Figure 3 gives the relationship between the threshold estimates and the likelihood ratio statistic, from which it can be seen that the threshold estimate of DE and EC is 0.2020 with a confidence interval of [0.1981, 0.2025], where the value of the likelihood ratio statistic is less than the critical value at the 5% level. Thus, the threshold effect estimate is considered to be true.

Based on the threshold test, we need to estimate the threshold model of DE on EC further, as shown in Table 8.

As can be seen from Table 8, the coefficient of GF on EC is 0.6178 and significantly positive at the 10% level when the DE does not exceed 0.2020, which indicates that GF has a significant contribution to EC when the DE level does not exceed the threshold value of 0.2020. The coefficient of GF on EC is −0.6190 and significant at the 1% level when the DE level exceeds 0.2020, which indicates that GF has a significant inhibitory effect on EC when the DE level exceeds the threshold value of 0.2020. To sum up these results, DE has a threshold effect on EC. For less developed regions of digital economy, such as Guizhou, Yunnan, and Shanxi, we should improve the construction of digital economy infrastructure and promote the development of DE, while also promoting the development of GF, promoting the upgrading of TC and EC, and promoting the development of GTFP. For regions such as Beijing, Guangdong, Anhui, etc., the infrastructure construction of DE is nearly perfect, and there are already good results for enterprise efficiency improvement. The inflow of GF funds to enterprises does not achieve good results for enterprise efficiency improvement, at which time we should restrict the inflow of GF funds to the efficiency improvement aspect of enterprises and encourage the flow of GF funds to aspects of technological progress, such as the research and development of independent intellectual property rights and the introduction of advanced equipment, so as to promote the continuous progress of GTFP in China.

We show no significant change in the number of thresholds and threshold coefficients by replacing the control variables section, which indicates that the threshold regression results are robust.

## 5. Conclusions

This paper empirically examined the impact and mechanism of green finance and the digital economy on GTFP, based on relevant data from 30 provinces, municipalities, and autonomous regions in China from 2006–2019. It was found that green finance and the digital economy have a significantly positive effect on GTFP, but they did not have a good effect when they acted together with GTFP. Based on this, a panel threshold model was introduced to investigate why the joint effects of green finance and the digital economy did not have a good effect on GTFP in depth. This paper hopes to provide a research basis for studies in related fields and provide a reference for government policy formulation.

Our study finds that:

(1)GF has a significant impact on green GTFP, with a coefficient of 3.25. Combined with existing studies, the mechanism is that green finance provides financial support to green industries, promotes the upgrading of industrial structure, fosters technological innovation, optimizes mineral resources for clean energy [58], and thus promotes GTFP. DE has a significant impact on GTFP, with a coefficient of 3.14. Combined with existing research, the mechanism is to optimize resource allocation by means of digital technology and promote the transformation and upgrading of industrial structure to digitalization, rationalization and greening, thus promoting GTFP.(2)The coefficient of the effect of GF acting on the GTFP decomposition term, TC, was 2.29, which was significant at the 5% level. The coefficient of the effect of GF acting on the GTFP decomposition term, EC, was 1.87, which was significant at the 10% level. The coefficient of the effect of DE acting on the GTFP decomposition term, TC, was 6.07, which was significant at the 1% level. The coefficient of the effect of DE acting on the GTFP decomposition term, EC, was −1.85, which was significant at the 10% level. Combining the available studies with this trend chart, GF mainly drives GTFP for the advancement of TC, and DE mainly drives GTFP for the advancement of TC.(3)When GF and DE acted together on GTFP, the effect was not significant. The “threshold effect” test reveals that there is a single threshold effect when GF and DE act together on EC. The threshold estimate is 0.2020, and the confidence interval is [0.1981, 0.2025], in which the likelihood ratio statistic is less than the critical value at the 5% level, and the threshold effect estimate is true. For regions with developed DE, the digital infrastructure is better built, and the productivity level of enterprises is relatively high when the GF funds acting on the EC side cannot produce good results. For the less developed areas of DE, the digital infrastructure construction has not yet reached a comprehensive level of perfection, and GF funds are able to flow into the enterprises to help improve the digital infrastructure construction, thus improving the level of enterprise productivity. At this time, GF funds can have a better impact on EC.

## 6. Policy Recommendations

Based on the above findings and analysis, this paper explores the following aspects of GF and DE to promote GTFP and makes the following recommendations:(1)Enhance the ability of green finance to drive green total factor productivity development.

In terms of green finance, financial institutions with green credit businesses should strengthen the support of green credit and establish a special green department to promote the development of green finance. Actively promote green investment business, accelerate the innovation of green financial products, expand green financial channels, encourage the participation of financial institutions and related enterprises, and disclose the development of green business. There should be the issuance of green credit guidelines and credit policies to provide credit support to green industries, such as photovoltaics, energy conservation, environmental protection, and new energy vehicles. The government should play an active role in forming an organic unity with green credit as the starting point and green investment and green insurance developing together to promote the progress of GTFP effectively.

(2)Improve the construction of digital infrastructure.

In terms of the digital economy, first of all, the green value of the digital economy should be fully explored, cross-regional allocation of digital resources should be promoted, the rapid development of 5G projects should be accelerated, and the business environment in each region should be optimized. Secondly, digital economy empowerment relies on institutional innovation related to the development of the digital economy and the introduction and improvement of laws and regulations, such as the Personal Information Protection Law and the Data Security Law, should be completed as soon as possible to explore the protection of intellectual property rights and personal privacy data security in the digital economy, as well as broaden the space for the development and the promotion of the quality of the digital economy. Finally, it is necessary to strengthen the construction of digital talents, and relevant universities and research institutes should open digital economy majors as soon as possible to enhance the effectiveness of training-related talents, which will promote the high-quality development of the digital economy.

(3)Pay attention to the driving effect of technological progress on green total factor productivity.

Technological progress is the main driving force for green total factor productivity improvement. Therefore, policymakers need to improve laws and regulations on the protection of independent intellectual property rights research and development to ensure that new technologies developed to promote green development are protected by law and to encourage enterprises to actively apply technologies related to green development in their business operations. In addition, the funds financed by enterprises through green finance channels should be supervised accordingly to ensure that these funds are applied to green production and operation projects of enterprises, which effectively promote the technological progress of enterprises and thus promote the development of GTFP.

(4)Implement differentiated green financial policies.

Focus on the “threshold effect” in the development of green finance and digital economy, and implement differentiated green finance policies according to local conditions and scientific guidance. For the less developed areas of China’s digital economy, policymakers should promote the development of green finance and the digital economy to promote GTFP development levels by driving technological progress and enterprise efficiency improvements. For the developed regions of China’s digital economy, policymakers should restrict the flow of green financial funds to the improvement of enterprise efficiency, avoid the blind flow of funds and disorderly development, and actively guide the flow of green financial funds to the research and development of independent intellectual property rights, the introduction of advanced equipment, industrial structure upgrading and other technological progress, so as to promote the green development of China’s economy.

## Figures and Tables

**Figure 1 ijerph-19-16303-f001:**
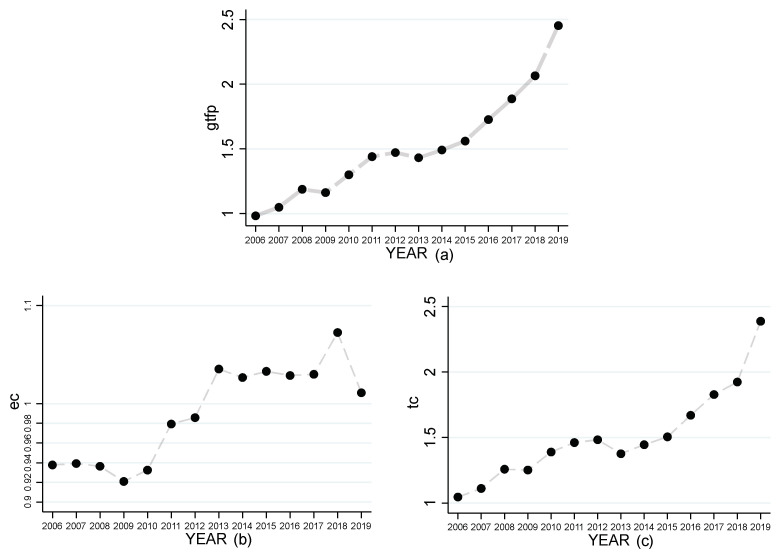
Technical efficiency change, Technical progress change and GTFP trend graphs. (**a**) indicates the trend of GTFP; (**b**) indicates the trend of technical efficiency change; (**c**) indicates the trend of technical progress change.

**Figure 2 ijerph-19-16303-f002:**
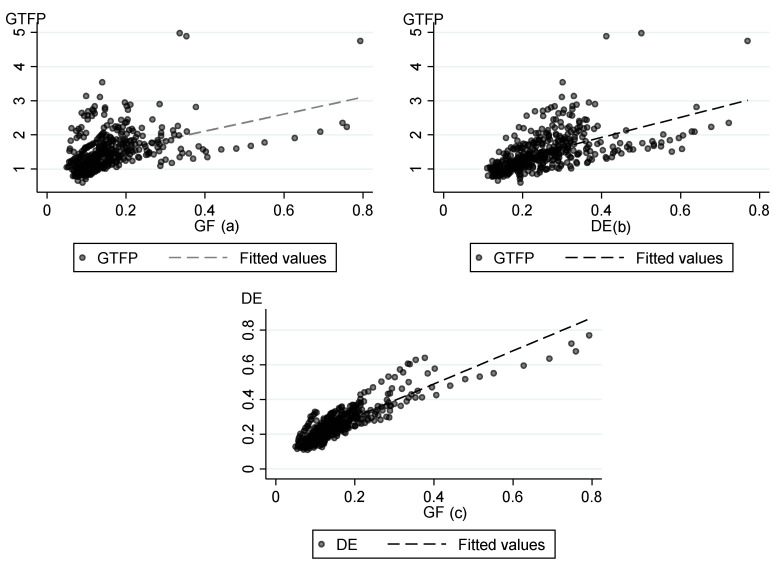
The relationship between GF, DE and GTFP. (**a**) shows the relationship between GF and GTFP; (**b**) shows the relationship between DE and GTFP; (**c**) shows the relationship between *GF* and DE.

**Figure 3 ijerph-19-16303-f003:**
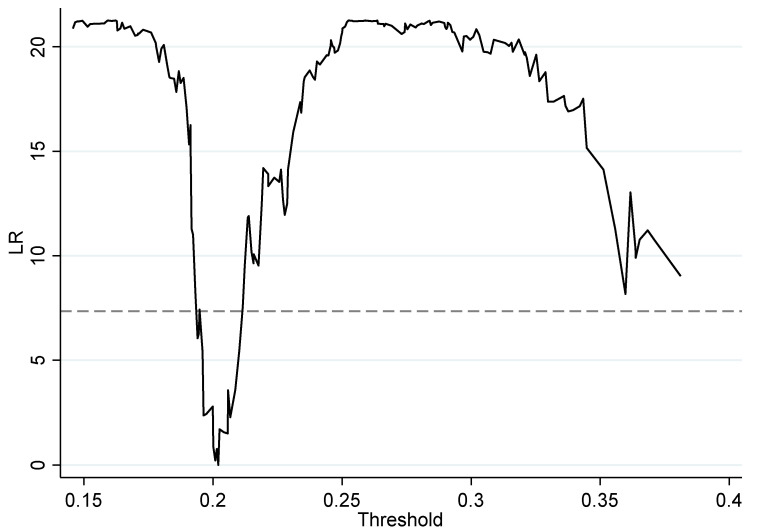
Threshold estimates and confidence intervals.

**Table 1 ijerph-19-16303-t001:** Green Finance Index.

Tier 1 Indicators	Characterization Indicators	Indicator Description
Green credit	Percentage of interest expenses in high-energy-consuming industries	Six high-energy-consuming industrial industries’ interest expenses/total industrial interest expenses
Green investment	Investment in environmental pollution control as a percentage of GDP	Environmental pollution control investment/GDP
Green insurance	Agricultural insurance depth	Agricultural insurance income/total agricultural output
Government support	Percentage of financial expenditure on environmental protection	Financial expenditure on environmental protection/financial general budget expenditure

**Table 2 ijerph-19-16303-t002:** Digital Economy Development Index.

Tier 1 Indicators	Secondary Indicators
Digital infrastructure	Long-distance cable density; Mobile phone switch capacity per capita; Number of internet ports per capita
Digital penetration rate	Internet penetration; Cell phone penetration; Number of websites per capita
	Number of websites owned by unit companies; Express business volume per capita
Digital technology talent benefits	Information transmission; Software and electronic technology service industry employees
	The proportion of employees in information transmission, software, and electronic technology services
	The number of legal entities in the information transmission, software, and electronic technology services industry
	Software business revenue; The number of resident populations at the end of the year
	Software business revenue per capita; Software business revenue as a percentage of GDP
	Total telecommunication services; Total telecom services per capita;
	Total telecom business as a proportion of GDP
	Electronic information manufacturing industry’s main business income actual value
	Actual value of main business income of electronic information manufacturing industry per capita
	Electronic information manufacturing industry’s main business income as a proportion of GDP
Digital research	Human capital; Education level; Education Funding; Education level
	Number of patent applications; Number of patent applications per capita

**Table 3 ijerph-19-16303-t003:** Descriptive statistics.

VarName	Obs	Mean	SD	Min	Max
GTFP	420	1.514	0.573	0.6080	4.9789
GF	420	0.160	0.099	0.0500	0.7930
DE	420	0.260	0.108	0.1104	0.7695
OIS	420	45.494	8.537	16.2000	61.5000
OPEN	420	0.022	0.020	0.0001	0.1210
URB	420	0.546	0.136	0.2746	0.8960
RD	420	0.010	0.006	0.0000	0.0324
GOV	420	3721.738	2686.780	174.54	17,297.85

**Table 4 ijerph-19-16303-t004:** Correlation analyses.

	*GTFP*	*GF*	*DE*	*OIS*	*OPEN*	*URB*	*RD*	*GOV*
GTFP	1							
GF	0.430 ***	1						
DE	0.554 ***	0.877 ***	1					
OIS	0.224 ***	0.444 ***	0.415 ***	1				
OPEN	0.002	0.159 ***	0.012	0.080	1			
URB	0.375 ***	0.591 ***	0.574 ***	0.336 ***	0.320 ***	1		
RD	0.312 ***	0.432 ***	0.452 ***	0.150 ***	0.371 ***	0.649 ***	1	
GOV	−0.078	0.243 ***	0.244 ***	−0.037	0.304 ***	0.390 ***	0.320 ***	1

Note: *** means *p* < 0.01.

**Table 5 ijerph-19-16303-t005:** LLC test results.

VarName	Statistic	*p*-Value
GTFP	−11.2467	0.0000
GF	−9.3897	0.0000
DE	−12.2336	0.0000

**Table 6 ijerph-19-16303-t006:** Multiple regression results.

	(1)	(2)	(3)	(4)	(5)	(6)
	*GTFP*	*TC*	*EC*	*GTFP*	*TC*	*EC*
*GF*	2.115 ***	1.011 **	0.434 *	1.885	−2.719 ***	2.687 ***
	(3.25)	(2.29)	(1.87)	(1.32)	(−2.88)	(5.48)
*DE*	1.991 ***	3.000 ***	−0.416 *	1.930 ***	2.005 ***	0.184
	(3.14)	(6.97)	(−1.85)	(2.68)	(4.21)	(0.75)
*OIS*	−0.008 **	−0.008 ***	0.001	−0.008 **	−0.011 ***	0.003 *
	(−2.07)	(−3.20)	(0.93)	(−2.07)	(−4.07)	(1.92)
*OPEN*	−4.462 ***	−1.911 *	−0.256	−4.536 ***	−3.099 ***	0.461
	(−3.05)	(−1.93)	(−0.49)	(−2.98)	(−3.08)	(0.88)
*URB*	3.125 ***	1.398 ***	1.156 ***	3.214 ***	2.841 ***	0.284
	(5.63)	(3.72)	(5.86)	(4.33)	(5.80)	(1.12)
*RD*	−40.160 ***	−36.677 ***	−0.966	−39.665 ***	−28.658 ***	−5.808 **
	(−4.96)	(−6.68)	(−0.34)	(−4.64)	(−5.07)	(−1.98)
*GOV*	0.000	−0.000	0.000	0.000	−0.000	0.000
	(0.11)	(−0.24)	(0.42)	(0.11)	(−0.21)	(0.40)
*GF × DE*				0.305	4.940 ***	−2.982 ***
				(0.18)	(4.45)	(−5.16)
_cons	−0.192	0.596 ***	0.351 ***	−0.200	0.460 **	0.433 ***
	(−0.62)	(2.86)	(3.21)	(−0.64)	(2.24)	(4.05)
N	420	420	420	420	420	420
r2	0.672	0.762	0.186	0.672	0.774	0.239

Note 1: The *t* statistics are in parentheses; * means *p* < 0.1, ** means *p* < 0.05, and *** means *p* < 0.01. Note 2: Due to the length of this article, only the double regression results are shown here.

**Table 7 ijerph-19-16303-t007:** Threshold effect test.

Core Explanatory Variables	Threshold Variables	Explained Variables	Models	Fstat	Prob	Crit1	Crit5	Crit10
*DE*	*GF*	*GTFP*	Single threshold	7.83	0.6533	31.1242	22.7221	19.7758
		*TC*	Single threshold	11.03	0.6533	43.2799	27.9036	24.6994
		*EC*	Single threshold	15.22	0.3033	41.3621	30.4086	24.0252
*GF*	*DE*	*GTFP*	Single threshold	6.80	0.6467	37.3207	23.3565	19.7139
		*TC*	Single threshold	16.23	0.4300	40.7447	30.5658	25.9309
		*EC*	Single threshold	27.46	0.0367 **	31.8307	22.9264	19.6111
			Double threshold	16.21	0.1033	33.2990	21.249	18.7108

Note: ** means *p* < 0.05.

**Table 8 ijerph-19-16303-t008:** Results of *DE* regression estimation on *EC* panel threshold model.

Variable Name		Coefficient	*t*	*p* > |*t*|	[95% Conf.	Interval]
*GF*	*EC* < 0.2020	0.6178002	1.65	0.100 *	−0.11998	1.35558
*GF*	*EC* > 0.2020	−0.6190305	−3.17	0.002 ***	−1.00325	−0.23481
*OIS*		0. 0034252	2.73	0.007 ***	0.00010	0.00590
*OPEN*		0.0611079	0.12	0.902	−0.91623	1.03845
*URB*		0.813084	4.52	0.000 ***	0.45942	1.16675
*RD*		−3.97947	−1.44	0.150	−9.40768	1.44874
*GOV*		1.46 × 10^−7^	0.06	0.953	−4.68 × 10^−6^	4.97 × 10^−6^

Note: The *t* statistics are in parentheses; * means *p* < 0.1 and *** means *p* < 0.01.

## Data Availability

Not applicable.

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
