# Peer review of "Exploring the Mechanism of the Impact of Green Finance and Digital Economy on China’s Green Total Factor Productivity"

_ijerph, 2022, doi:10.3390/ijerph192316303_

Round 1

Reviewer 1 Report

Comments to the authors

The manuscript entitled "Exploring the mechanism of the impact of green finance and digital economy on China's total green factor productivity" can contribute significantly to the existing literature concerning sustainable development goals. This article has scientific merit for publication but requires major revision before publication in this journal. My suggestions are as follows: 

#in the abstract, there are some grammatical errors. Please correct it.

#the name of the methodology should be mentioned in the abstract.

# in this sentence in the abstract, "Therefore, we should formulate differentiated green financial policies", the authors must say ‘policymakers’ but not 'we'. 

#the authors must write the study's contribution, methodology and results in brief in the introduction. The introduction is too small.

#the authors must add a literature review section and develop the hypothesis based on the literature reviewed. In this regard, please follow these papers:
https://www.sciencedirect.com/science/article/pii/S0140988322003541

https://www.sciencedirect.com/science/article/abs/pii/S0160791X22002731?via%3Dihub https://www.sciencedirect.com/science/article/pii/S0140988322002341?via%3Dihub

#section 2.1.2 should be displaced from this existing section to the literature review section.

#data descriptions are too large. Please make it short.

#Legend should be inserted beneath all the tables added to the manuscript.

#in the result and discussion section, please add supportive and opposite works of literature. Moreover, this section requires to be improved and expanded.

#conclusion and policy implications are not properly written. Please represent this scientifically and systematically. In this regard, please follow this paper and cite it: https://www.sciencedirect.com/science/article/abs/pii/S0301420722003385

Good luck!  

Author Response

The manuscript entitled "Exploring the mechanism of the impact of green finance and digital economy on China's total green factor productivity" can contribute significantly to the existing literature concerning sustainable development goals. This article has scientific merit for publication but requires major revision before publication in this journal. My suggestions are as follows: 

#in the abstract, there are some grammatical errors. Please correct it.

Response: Thank you very much for your valuable comment and suggestion, we checked the abstract for grammatical problems and made changes as recommended.

#the name of the methodology should be mentioned in the abstract.

Response: Thank you very much for your valuable comment and suggestion, we mention the methods used in this paper in the abstract section as recommended please verify it.

# in this sentence in the abstract, "Therefore, we should formulate differentiated green financial policies", the authors must say ‘policymakers’ but not 'we'. 

Response: Thank you very much for your valuable comment and suggestion, we changed the ‘we’ in the abstract to ‘policymakers’ as recommended please verify it.

#the authors must write the study's contribution, methodology and results in brief in the introduction. The introduction is too small.

Response: Thank you very much for your valuable comment and suggestion, we have expanded the introduction and added the study's contribution, methodology and results to the introduction as recommended please verify it.

#the authors must add a literature review section and develop the hypothesis based on the literature reviewed. In this regard, please follow these papers:
https://www.sciencedirect.com/science/article/pii/S0140988322003541
https://www.sciencedirect.com/science/article/abs/pii/S0160791X22002731?via%3Dihub https://www.sciencedirect.com/science/article/pii/S0140988322002341?via%3Dihub

Response: Thank you very much for your valuable comment and suggestion, we have read and cited the literature you provided, added a literature review section and discussed existing studies as recommended please verify it.

#section 2.1.2 should be displaced from this existing section to the literature review section.

Response: Thank you very much for your valuable comment and suggestion, we removed section 2.1.2 and moved the valuable content to the literature review section as recommended please verify it.

#data descriptions are too large. Please make it short.

Response: Thank you very much for your valuable comment and suggestion, we narrowed down the data description section as recommended.

#Legend should be inserted beneath all the tables added to the manuscript.

Response: Thank you very much for your valuable comment and suggestion, we have inserted note beneath all tables as recommended please verify it.

#in the result and discussion section, please add supportive and opposite works of literature. Moreover, this section requires to be improved and expanded.

Response: Thank you very much for your valuable comment and suggestion, we added a comparison with existing research literature and upgraded and expanded the results and discussion section as recommended please verify it.

#conclusion and policy implications are not properly written. Please represent this scientifically and systematically. In this regard, please follow this paper and cite it: https://www.sciencedirect.com/science/article/abs/pii/S0301420722003385

Response: Thank you very much for your valuable comment and suggestion, we have scientifically and systematically revised the conclusions and policy implications section by referring to and citing the literature you provided.

Reviewer 2 Report

1.The authors chosen threshold regression try to explain the contribution of green finance to efficiency improvement in the some regions of the digital economy. It is a brand new thinking, and chosen methodology is comprehension.

2.In this study, just including the data which from 2006 to 2019, which was skip some major impact in China market (like SARS in 2003, covid19 and so on), it is may let some bias for conclusions in this study. And the data of this study is too “dated”. Have to renew all of data of research.

3. The  threshold regression adopted in this study determines the threshold value according to the information of the samples themselves. However, given that the parameter estimation in this research method is an inference made under the assumption that the variance is consistent, the correctness of some parameter estimation may be affected when there is heterogeneity in the data. Therefore, it is suggested to make correction for heterogeneity.

4.It is suggested that the author should ask the MDPI editing center to revise the paper and submit it again.

Reviewer 3 Report

The manuscript uses appropriate method on an economic topic of regional interest. The writing is generally acceptable, but there are two issues: the abstract and the introduction (and some of the following review). The abstract should be re-written, and the introduction should be focused on the paper rather than on only mildly interesting general discussion.

Details:

abstract, l.14: "The interrelationship ... was empirically tested" (suggested)

abstract, l.20: "show a stronger contribution" (suggested)

l.99: "enterprises that want to close" ?? It appears that authors want to discuss enterprises that want to prodce, not those that want to close their businesses

l.184: the abbreviation ML should be convened

l.217: "economics human" ??

l.234: "deep" used as a verb: unclear

l.277: space below table, this is not a table footnote

l.296: end sentence here, start a new one "China's economic transformation advocates the concept" end it with "protection"

l.298 start a new sentence "For this reason, high-pollution enterprises"

l.299 verb missing "and thus are more willing"

Table 5: replace decimal commas by decimal points. Replace 420,000 by 420

l.424; not 5% confidence but 5% significance

l.448: "relatively perfect" unclear, probably nearly perfect

l.475: "significant" omit or "strong" (confused with statistical usage)

l.518,522,527: "we" appears to be China, which is not adequate for an academic publication. "we" should refer to authors ("we find, we suggest") not to authors+reader (we see in the table) and not to a nation. Use abstract entities such as "policy"

Author Response

The manuscript uses appropriate method on an economic topic of regional interest. The writing is generally acceptable, but there are two issues: the abstract and the introduction (and some of the following review). The abstract should be re-written, and the introduction should be focused on the paper rather than on only mildly interesting general discussion.

Response: Thank you very much for your valuable comment and suggestion. We have rewritten the abstract section to include the results of the methods used for the study in the abstract section. We have also rewritten the introduction section to include the methodological results of the study and the contribution of this paper.

Details:

abstract, l.14: "The interrelationship ... was empirically tested" (suggested)

Response: Thank you very much for your valuable comment and suggestion, we have included the methodology used for the study and written the results of the empirical tests here as recommended.

abstract, l.20: "show a stronger contribution" (suggested)

Response: Thank you very much for your valuable comment and suggestion, we have rewritten this sentence as recommended.

l.99: "enterprises that want to close" ?? It appears that authors want to discuss enterprises that want to prodce, not those that want to close their businesses

Response: Thank you very much for your valuable comment and suggestion, we have rewritten this sentence as recommended please verify it.

l.184: the abbreviation ML should be convened

Response: Thank you very much for your valuable comment and suggestion, we have added the ML abbreviation here as recommended please verify it.

l.217: "economics human" ??

Response: Thank you very much for your valuable comment and suggestion, we have rewritten this sentence as recommended please verify it.

l.234: "deep" used as a verb: unclear

Response: Thank you very much for your valuable comment and suggestion, we have rewritten this sentence as recommended please verify it.

l.277: space below table, this is not a table footnote

Response: Thank you very much for your valuable comment and suggestion, we have added space here as recommended please verify it.

l.296: end sentence here, start a new one "China's economic transformation advocates the concept" end it with "protection"

Response: Thank you very much for your valuable comment and suggestion, we have adjusted the sentence format here as recommended please verify it.

l.298 start a new sentence "For this reason, high-pollution enterprises"

Response: Thank you very much for your valuable comment and suggestion, we have split the sentence here as recommended please verify it.

l.299 verb missing "and thus are more willing"

Response: Thank you very much for your valuable comment and suggestion, we have reworked the sentence here as recommended.

Table 5: replace decimal commas by decimal points. Replace 420,000 by 420

Response: Thank you very much for your valuable comment and suggestion, we changed the comma to a decimal point as recommended please verify it.

l.424; not 5% confidence but 5% significance

Response: Thank you very much for your valuable comment and suggestion, we changed confidence to significance as recommended please verify it.

l.448: "relatively perfect" unclear, probably nearly perfect

Response: Thank you very much for your valuable comment and suggestion, we changed relatively perfect to nearly perfect as recommended please verify it.

l.475: "significant" omit or "strong" (confused with statistical usage)

Response: Thank you very much for your valuable comment and suggestion, we deleted significant as recommended please verify it.

l.518,522,527: "we" appears to be China, which is not adequate for an academic publication. "we" should refer to authors ("we find, we suggest") not to authors+reader (we see in the table) and not to a nation. Use abstract entities such as "policy"

Response: Thank you very much for your valuable comment and suggestion, we changed the ‘we’ in these sentences to ‘policymakers’ as recommended please verify it.

Reviewer 4 Report

Referee report

Main comments:

The abstract should define the concepts of green finance, digital economy, and green total factor productivity. Especially how they are defined in the data for the regression analysis. Lines 82-87 on page 2 do not provide a clear definition either.

The paper should explain the correctness of using the data envelope analysis for obtaining the measure of green total factor productivity. Why the choice of 9.6% for the depreciation rate is considered? Why investment is considered to be fixed in the panel data analysis? Equations 1-3 are not clearly described.

What is the composite index method for creating the green finance index?

Why do green finance and digital economy jointly have not a good effect on green total factor productivity? What do “a good effect” and “not a good effect” mean?

I am not sure if replacing the missing data by approximated values based on linear interpolation is adequate. Probably one would need to analyze the data where the missing observations are excluded.

How do equations 5-6 relate to this paper’s topic? The time fixed effects, individual fixed effects, and error terms cannot be the same in all three equations.

For testing the nonlinearity, panel Ramsey test should have been performed before deciding to add the interaction terms. Some of variables seem to be nonstationary. Why stationarity is not tested? How is nonstationarity resolved?

I think that reviewing results is not relevant at this stage because I have serious comments related to the methodology.

Minor comments:

Lines 126-128 on page 3 and lines 226-231 on page 6 should be rewritten.

A list of abbreviation should be added.

Author Response

The abstract should define the concepts of green finance, digital economy, and green total factor productivity. Especially how they are defined in the data for the regression analysis. Lines 82-87 on page 2 do not provide a clear definition either.

Response: Thank you very much for your valuable comment and suggestion, we have rewritten the introduction and added the definitions of green finance, digital economy and green total factor productivity to the introduction as recommended.

The paper should explain the correctness of using the data envelope analysis for obtaining the measure of green total factor productivity. Why the choice of 9.6% for the depreciation rate is considered? Why investment is considered to be fixed in the panel data analysis? Equations 1-3 are not clearly described.

Response: Thank you very much for your valuable comment and suggestion. We add Data Envelopment Analysis (DEA) to the article to measure the correctness of green total factor productivity. Since Data Envelopment Analysis (DEA) has the advantage of not requiring functional assumptions, and the non-angle, non-radial distance Malmquist index (ML) can treat pollution emissions as non-desired output and solve the problem of radial distance function, it can achieve a decrease in non-desired output and an increase in desired output at the same time. Regarding the fixed asset investment 9.6% is currently the accepted depreciation rate in Chinese academia, we choose 9.6% as the fixed asset depreciation rate after comprehensive consideration. We have rewritten equations 1-3 as recommended.

What is the composite index method for creating the green finance index?

Response: Thank you very much for your valuable comment and suggestion, we add to the article the methodology for measuring green finance as recommended.

Why do green finance and digital economy jointly have not a good effect on green total factor productivity? What do “a good effect” and “not a good effect” mean?

Response: Thank you very much for your valuable comment and suggestion, we rewrote this part of our article explaining the impact of green finance and the digital economy on green total factor productivity. GF is able to produce significant positive effects when acting on GTFP as well as when EC acts on GTFP, but when GF is combined with DE it has an inhibitory effect on EC during production. Since GTFP=EC×TC, the joint effect of GF and DE shows a significant increase in TC but a significant inhibition in EC, so the joint effect of GF and DE does not significantly increase GTFP.

I am not sure if replacing the missing data by approximated values based on linear interpolation is adequate. Probably one would need to analyze the data where the missing observations are excluded.

Response: Thank you very much for your valuable comment and suggestion. The missing data are due to individual years in individual regions, and there are only two missing data, so it is more prudent for us to use linear interpolation method to deal with them.

How do equations 5-6 relate to this paper’s topic? The time fixed effects, individual fixed effects, and error terms cannot be the same in all three equations.

Response: Thank you very much for your valuable comment and suggestion, ML=EC×TC, ML cumulative multiplication to obtain GTFP. We have revised the relevant parts of the article. Equations (5) and (6) lie in the analysis of the effects on TC and EC. Due to the length of the article, the table regression results are shown only for the double fixed regression results. And we modify the footnote of the table to add an explanation for this reason.

For testing the nonlinearity, panel Ramsey test should have been performed before deciding to add the interaction terms. Some of variables seem to be nonstationary. Why stationarity is not tested? How is nonstationarity resolved?

Response: Thank you very much for your valuable comment and suggestion, we added an LLC test for testing the smoothness of the data, and the results showed that the data were smooth and the results were added to the article as recommended.

Minor comments:

Lines 126-128 on page 3 and lines 226-231 on page 6 should be rewritten.

Response: Thank you very much for your valuable comment and suggestion, we have rewritten these sentences as recommended please verify it.

A list of abbreviation should be added.

Response: Thank you very much for your valuable comment and suggestion, we have added abbreviations to the first appearance of proper nouns as recommended please verify it.

Round 2

Reviewer 1 Report

Authors have addressed the issues raised by me. Thanks for your hard efforts! Good luck!